# Agency Contracts under Maximum-Entropy

**DOI:** 10.3390/e23080957

**Published:** 2021-07-26

**Authors:** Oscar Gutiérrez, Vicente Salas-Fumás

**Affiliations:** 1Department of Business Economics, Universitat Autònoma de Barcelona, 08193 Barcelona, Spain; 2Departamento de Economía y Dirección de Empresas Universidad de Zaragoza, 50005 Zaragoza, Spain; vsalas@unizar.es

**Keywords:** maximum-entropy, agency relationship, moral hazard, first-order approach, likelihood ratio, worst-case scenario

## Abstract

This article proposes the application of the maximum-entropy principle (MEP) to agency contracting (where a *principal* hires an *agent* to make decisions on their behalf) in situations where the principal and agent only have partial knowledge on the probability distribution of the output conditioned on the agent’s actions. The paper characterizes the second-best agency contract from a maximum entropy distribution (MED) obtained from applying the MEP to the agency situation consistently with the information available. We show that, with the minimum shared information about the output distribution for the agency relationship to take place, the second-best compensation contract is (a monotone transformation of) an increasing affine function of output. With additional information on the output distribution, the second-best optimal contracts *can be* more complex. The second-best contracts obtained theoretically from the MEP cover many compensation schemes observed in real agency relationships.

## 1. Introduction

The concept of entropy is central in information theory, but ignored in the economics of information, an area of research in economic theory interested in the valuation of information and knowledge as productive resources. From the seminal paper [1], where information is a product of “search” by buyers and sellers in the market, the field has evolved into the non-cooperative game theoretic analysis of the role of information in contractual relationships where parties have different information in the contracting stage and/or in the contract implementation stage (see [2]).

This article proposes a first analysis in closing this gap. We appeal to the maximum-entropy principle to analyze the economic problem of optimal contracting in agency relationships, in which a *principal* hires an *agent* to make decisions on their behalf. When these decisions are unobservable for the principal (information asymmetry in the contract implementation stage), the agent may behave opportunistically, which is known as a *moral hazard*. This problem, central in microeconomics and information economics, has attracted a lot of research (to cite only some Nobel laureates who have contributed to agency theory, we can mention B. Holmstrom, J. Mirrlees, K. Arrow, E. Maskin, P. Diamond, O. Hart, P. Milgrom, M. Spence, E. Fama or J. Stiglitz.) The present analysis invokes the maximum-entropy principle to model and solve for optimal contracts in agency relationships under moral hazard when the available information for the contracting parties is scarce (Ref. [3] pioneered the application of *entropy* to economics. In the last years, the concept of entropy has been increasingly used in economics and finance; see [4,5,6,7] for reviews). The application of the maximum-entropy principle to decision-making in agency relationships is novel, but wholly natural considering the central role that information plays in the design of optimal agency contracts. The concept of entropy has been extensively applied to (physical, biological, economic) systems with many degrees of freedom, and also to econometrics and time series analysis (where data is, to more or less extent, abundant); see [8,9,10]. In contrast, the approach to agency based on the maximum-entropy principle deals with a contracting problem where information is extremely scarce, so the present approach is conceptually closer to the maximum-entropy principle in the formulation of [11], than to the more classical concept of entropy as a measure of disorder. To the best of our knowledge, ref. [12] (ft. 5) is the only reference in which the concept of *entropy* appears in the context of managerial compensation contracts

In the solution to the standard agency problem, it is generally assumed that principal and agent both know and agree on the probability output distribution (conditioned on the agent’s action); see [13] (p. 76). However, how this agreement takes place has not been investigated thus far. This article proposes the *maximum entropy principle* (MEP henceforth) to investigate agency contracting in situations of partial information about the output distribution. By adopting the MEP, principal and agent agree in restricting the contracts to those compatible with the available information they share.

In individual decision-making under uncertainty, the MEP provides a criterion to choose a probability distribution consistent with the information that the decision-maker has at the time of the decision, therefore avoiding biases of making decisions based on information that is non-available; see [11,14,15]. The probability distribution selected under this criterion satisfies [16] conditions for any uncertainty measure: being continuous, symmetric, and additive for independent sources of uncertainty. Under the MEP, the probabilities of the states of nature incorporate the information that the decision-maker has at the time of the decision, rather than representing frequencies of past realizations of the random variable. This makes the approach compatible with the “subjective school” on the nature of probability (see [17]): the probability of an event is not an objective property of that event, but a mere expression of the human ignorance (see [11] (p. 622)). For these reasons, the MEP represents a coherent response to decision situations under “Knightian uncertainty” ([18]). The distribution coming out of the application of the MEP is called *maximum-entropy distribution* (MED henceforth). By construction, it is the least informative distribution ex-ante and the most informative distribution ex-post, when the realization of the random variable has been observed. It is also the least biased estimate of the unknown distribution given the prior information, avoids unconsciously incorporating arbitrary assumptions in the chosen distribution, and any event that cannot be excluded to occur will have a positive probability.

Much less is known about the application of the MEP to contracting situations between two (or more) parties in situations of asymmetric information, as is the case in agency contracting. In agency relations, the principal, beneficiary of the agent’s work, observes the output (which is influenced by the agent) but not their actions (or effort). In a traditional agency, to solve for the second-best optimal contract, it is assumed that the principal and agent agree on the ex-ante distribution of output (conditioned on the agent’s effort), but the conditions under which such agreement is reached have not been investigated thus far. This article proposes the MEP as an appropriate framework to examine the problem when the output distribution is unknown, the principal and agent only share partial information about it, and the agreement around which distribution to use must conform with the existing shared information, not more not less. The application implies first finding a meaningful way to incorporate the decision (action or effort) of the agent in the optimization problem that will give the output MED (conditioned on agent’s effort) and, then derive the second-best optimal contracts (consistently with the information embedded in the maximum-entropy problem with constraints).

The results of the analysis link the information shared by the principal and agent with the characteristics of the second-best optimal contract. Intuitively, scarce information results in simpler second-best compensation contracts (in particular, if the only shared information is that the agent’s effort drives the expected output, the second-best contract is transformed-affine in output). With additional shared information, more complex compensation contracts (convex or concave with output) can be, theoretically, second-best optimal (if, for example, the agent’s effort affects the mean and the variance of the output distribution, the second-best contract can be convex in output). Then, the article compares the characteristics of the second-best contracts characterized from the application of the MEP with commonly used managerial compensation contracts (see [19,20,21] (Ch. 7)), one of the areas where agency theory has demonstrated its more practical use.

The rest of the article is structured as follows. In Section 2, we present the basic version of the framework (discrete output). In Section 3, we use the first-order approach to the principal-agent model to show that the optimal contract is affine in a utility-transformed space. Some extensions are presented in Section 4: the framework is generalized to continuous output and the incorporation of additional pieces of information is allowed; this section also includes a numerical example that illustrates an application of the model. Section 5 discusses the results, and Section 6 concludes.

## 2. Agency, Information and Maximum-Entropy

### 2.1. Assumptions

This subsection presents and discusses the model assumptions. First, we introduce a common version of the agency setup, the standard principal–agent model under moral hazard formulated by [13]. Then, the model assumptions are introduced. The first one, (A1), is only an additional requirement made for mathematical tractability, usual in agency. (A2) and (A3) represent the specific assumptions of the MEP approach to agency in its basic version.

#### 2.1.1. Agency Setup and Assumptions

One person (*principal*) hires another person (*agent*) to perform a task on their behalf that will result in a monetary payoff for the principal according to the production function x=x(a,θ)*,*
(∂/∂a)x(a,θ)≥0, where *a* is the quantity of resource input contributed by the agent and *θ* is the realization of a random state of nature. The input has an opportunity cost for the agent, *C*(*a*), strictly increasing and weakly convex: *C′*(*a*) > 0, *C″*(*a*) ≥ 0. The agent’s input *a* is private information of the agent, but the principal and agent observe the output *x*. The principal offers the agent an output-dependent compensation *s*(*x*) for the costly input (effort); the net utility of the agent is U(w)−C(a), where *w* represents wealth (which in our case corresponds to *s*(*x*)) and U(·) is an strictly increasing and strictly concave von Neumann–Morgenstern utility function (risk-averse agent): $U^{\prime }(\cdot )>0$, $U^{\prime \prime }(\cdot )<0$. The basic model assumptions of the MEP approach are:

**Assumption** **1** **(A1).***F**unction**U*[*J*(·)], *with*$J(x):={\left(U^{\prime }\right)}^{-1}(1/x)$*is concave.*

**Assumption** **2** **(A2).***Output is**non**-negative, x ≥ 0, and**can adopt N discrete positive realizations,*$x\in \{x_{1},\ldots ,x_{N}\}$.

The expected output is driven by the agent’s effort. In particular, and for the sake of simplicity, but without loss of generality, E(X)=a. 

**Assumption** **3** **(A3).**
*Principal and agent converge to the information-constrained output distribution according to the Maximum-Entropy Principle.*


#### 2.1.2. Discussion of Assumptions

(A1) The concavity of *U*[*J*(·)], with $J(x):={\left(U^{\prime }\right)}^{-1}(1/x)$, is introduced in order to justify the application of the first-order approach to agency problems (see [22] (Equations (2.9) and (2.12))). The application of the first-order-approach simplifies the mathematical solution of the principal–agent contracting problem.

(A2) In business contexts, the non-negative output could be the sales of a salesperson or the market value of a firm managed by a professional CEO.

Agency theory has generally taken for granted that in the contracting stage, the principal and agent agree on the probability distribution *F*(*x*, *a*) on $x\in \{x_{1},\ldots ,x_{N}\}$ for a given *a*; see [13,23]. For a given input quantity *a,* each of the *N* possible realizations of states of nature yields an output value $x_{i}=x(a,\theta_{i})$, *i* = 1,…, *N*. Therefore, there is a correspondence between the random variable associated to the states of nature *θ* and the random variable of output *X*, which in our case satisfies *x* ≥ 0, $x\in \{x_{1},\ldots ,x_{N}\}$.

The MEP approach to agency proposed in this paper makes sense in situations where the principal and agent do not know the probability distribution. However, for the principal to be interested in contracting the agent, some minimum conditions are required. One of these (necessary) conditions is that the principal knows that *E*(*X*|*a*) ≥ *C*(*a*) for some values of *a*. This condition means that it is worth contracting the agent to perform the job because the expected output is higher than (or equal to) the opportunity cost of the agent’s effort. In our MEP framework, we assumed that *E*(*X*) = *a* (>*C*(*a*)). Assuming E(X)=a for *principal and agent* is innocuous because if the principal knows or conjectures that E(X)=a, the cost function *C*(*a*) can be re-scaled in such a way that the agent will agree on E(X)=a; and the agent could disagree with the principal about how the agent’s action influences on expected output *E*(*X*), but the notion of effort must be linked to the corresponding opportunity cost. This implies that the agent will be willing to accept E(X)=a whenever an adequate opportunity (or monetary) cost is associated to *a* (given their reservation utility), and the principal will consider acceptable the re-scaled opportunity cost whenever their expected profit is positive. (If E(X)=a is private information of the agent, we have a classical agency problem with adverse selection. By the revelation principle, the principal may offer a menu of contracts as a screening mechanism to elicit the agent’s information).

Assumption 2 (A2) represents a key feature of our approach; through the constraint E(X)=a, the agency relationship is embedded in the decision-making procedure proposed by the MEP. 

(A3) The entropy H of the probability distribution $\left\{p_{1},\ldots ,p_{N}\}\equiv \{\mathrm{Pr}(x_{1}),\ldots ,\mathrm{Pr}(x_{N})\right\}$ gives a measure of the uncertainty of the output distribution, and is defined as $H(p_{1},\ldots ,p_{N})\overset{def}{=}-\sum_{i=1}^{N}p_{i}\ln(p_{i})$. The MEP prescribes choosing the probability distribution with the maximum entropy (i.e., consistently with the information available and no more than that). This implies that if the principal and agent reason from the shared information under the information theory principles, the two will separately choose the same probability distribution. Note that the principal and agent do not necessarily explicitly agree to invoke the MEP to converge to an output distribution. Rather, the two face a situation of “shared ignorance”, in which they only share some pieces of information (in the simplest case the information reduces to *E*(*X*) = *a*; see (A2)), and individually follow the MEP to form their prior distributions. The two adhere to the MEP to make sure that the proposed output distribution does not incorporate information that they do not have.

Information theory (see [16]) proposes measuring the uncertainty faced by a decision-maker by the *entropy* of the probability distribution $\left(p_{1},\ldots ,p_{N}\right)$, defined above. A simple explanation for the functional form of the entropy given is as follows: if the event $x_{i}$, whose probability is $p_{i}$, is the realized one, then an amount of information $-\ln(p_{i})$ is gained. This specification ensures that: (i) the information is additive, and (ii) rarer events provide more information if they are the observed ones. Then, the information carried by a distribution will be equal to the average information contained in its associated events: $-\sum_{i=1}^{N}p_{i}\ln(p_{i})$. Other information measures could be used, for example $-\sum_{i=1}^{N}p_{i}^{2}$, but [11] (Section 2 and Appendix A) justifies why Shannon’s measure is the most adequate one, in order to ensure positive probabilities and logical consistency. It can be said that probability measures uncertainty about the occurrence of a single event, while entropy measures the uncertainty of a collection of events ([24] (p. 495)). It is worth observing that entropy and variance are different concepts: entropy refers to information and variance to risk. In the multivariate case, entropy and variance can be related (see [25] and references therein). From information theory, the maximum-entropy distribution is justified because it represents the only unbiased assignment based on the given information (i.e., the distribution is maximally noncommittal on missing information, see [11,16]). As a consequence, the entropy of the distribution can be seen as a measure of the information gained when uncertainty is unraveled. 

### 2.2. The Maximum-Entropy Distribution

From the application of Assumptions (A2) and (A3), the MED is that resulting by solving the following program:$$\underset{p_{1},\ldots ,p_{N}}{\max}H(p_{1},\ldots ,p_{N})$$
(P1)$$\mathrm{subject}\;\mathrm{to}:\;\sum_{i=1}^{N}p_{i}^{}=1$$
$$\mathrm{and}:\;E(x)=\sum_{i=1}^{N}p_{i}^{}x_{i}=a.$$

The solution to problem (P1) is straightforward (see [11]). The entropy-maximizing probability values correspond to a particular case of the softargmax functions:(1)$$p_{i}=e^{-A-Bx_{i}}$$
with *A* = ln*Z*(*B*), $Z(B)\equiv \sum_{i=1}^{N}e^{-Bx_{i}}$, and *B* obtained from $a=-\frac{\partial }{\partial B}\ln(Z(B))$.

The probabilities of the distributions that comply with the MEP fit the functional form given in Equation (1), with *A* and *B* the Lagrange multipliers of the two constraints in (P1). One important example is the geometric distribution (outcomes $\left\{x_{1},\ldots ,x_{N}\right\}$ are then equally spaced), and under particular circumstances the binomial distribution also belongs to the family of MEDs (see Proposition 2(a)). It is worth observing that the probabilities decrease with the value of the output (i.e., not concentrated around the modal value of the output variable). In fact, Equation (1) implies that if the only information shared by the principal and agent is *E*(*X*) = *a*, the ME approach does not generate bell-shaped (or inversely U-shaped) probability functions, similar to the Gibbs distribution for microstates. In Section 3, we return to this apparent limitation of the MEP approach to agency, showing how probability distributions with a bell-shaped pattern can arise from the MEP.

### 2.3. Relation with Game Theory

The MED has also been justified as the equilibrium solution of a zero-sum game that the decision maker (DM henceforth) plays against nature, when nature randomly chooses the probability distribution of the states of nature and the loss function of the DM is logarithmic (the so-called “log-loss” game). This choice of the loss function is not capricious, but closely related to the very notion of information introduced by [16], and is consistent with the rationale of entropy given in the discussion of (A3). The conclusion is robust to the choice of other mathematical specifications of the loss function; see [26] (Section 3). The study in [26] (Section 2.1) shows that in the log-loss game, the MED characterizes both nature’s max-min and DM’s min-max strategies of the game. As a consequence, reading “loss” as “the negative of utility”, if the DM selects a distribution P* by appealing to the MEP, the choice is compatible with the equilibrium solution of a max-min expected utility decision model. (Ref. [27] proposed a game-theory approach to agency models in an extension of the compensation model [28], in which both the principal and agent act as if “nature” is playing against them by choosing the worst possible volatility of the output.) The rationales from information theory and game theory for using maximum-entropy distributions are linked through the so-called Code Length Game, a zero sum game analyzed by [29] (Section 3).

## 3. The Second-Best Agency Contract

### 3.1. Basic Result

Having agreed on the MED, the risk-neutral principal solves for the contract that maximizes the expected pay-off *E*(*x* − *s*(*x*)), subject to the agent’s participation and incentive constraints:$$\underset{s(x),a}{\max}\sum_{i=1}^{N}\left[x_{i}-s(x_{i})\right]p_{i}(a)$$
(P2)$$\mathrm{subject}\;\mathrm{to}\;:\sum_{i=1}^{N}U(s(x_{i}))p_{i}(a)-C(a)\geq R$$
$$a\in \underset{a^{\prime }}{\arg\max}\sum_{i=1}^{N}U(s(x_{i}))p_{i}(a^{\prime })-C(a^{\prime })$$
where ${\left\{p_{i}(a)\right\}}_{i=1,\ldots ,N}$ represents the maximum-entropy output distribution given an effort equal to a, see Equation (1), and *R* is the reservation utility of the agent. The constraints in (P2) are respectively known as participation constraint and incentive compatibility constraint.

For the solution of problem (P2), the incentive compatibility constraint is (justifiably) substituted by the first-order condition, the so-called first-order-approach (FOA henceforth): $a\in \underset{a^{\prime }}{\arg\max}\sum_{i=1}^{N}U(s(x_{i}))p_{i}(a^{\prime })-C(a^{\prime })$ is replaced by the relaxed constraint $\sum_{i=1}^{N}U\left[s(x_{i})\right]p_{i}^{\prime }(a)-C^{\prime }(a)$, with $p_{i}^{\prime }(a)\equiv \partial p_{i}(a)/\partial a$; see [22,30] for seminal sufficient conditions of application of the FOA, and [31,32,33] for more recent research on this topic. From the first-order approach, the optimal contract *s**(*x*) satisfies the following two conditions:(2a)$$\frac{1}{U^{\prime }(s*(x))}=\lambda +\mu \frac{p_{i}^{\prime }(a*)}{p_{i}(a*)}$$
(2b)$$\sum_{i=1}^{N}\left[x_{i}-s(x_{i})\right]p_{i}^{\prime }(a)+\mu \left[\sum_{i=1}^{N}U(s*(x_{i}))p_{i}^{\prime \prime }(a*)-C^{\prime \prime }(a*)\right]=0$$
where *λ* > 0 and *μ* > 0 represent the Lagrange multipliers associated with the participation and the incentive compatibility constraints, respectively, and *a** is the second-best level of effort induced by the contract *s**(*x*) (when the FOA holds the Lagrange multipliers exist and are positive; see [22]). The optimal contract is then derived from (2a) as a function of *λ*, *μ*, and *a**, and these three unknowns are obtained from the two constraints in the relaxed version of (P2) together with (2b). The second-best agency contract in the maximum-entropy setup is given next.

**Proposition** **1.**
*If the output distribution is the solution to *(*P1*)*, then the second-best agency contract is a monotone transformation of an increasing affine function of output.*


**Proof.** It is shown that the second-best agency contract is “affine in a transformed space”, which in our context means that the agent’s compensation can be written as J(α+βx), with *α* and *β* real constants and J(·) being the monotone transformation of U(·) given in (A1), $J(x)={\left(U^{\prime }\right)}^{-1}(1/x)$.

The MED in (1) belongs to the exponential family. The validity of the FOA, which simplifies the solution to the second-best optimal agency contract, in our context, can then be shown to apply by appealing to [22] (Cor. 1) or to [33] (Cor. 4); recall that *E*(*X*) = *a* and that *U*[*J*(·)] is concave by Assumption 1 (A1). Then, the second-best contract is $s*(x)=J\left[\lambda +\mu \frac{p_{i}^{\prime }(a*)}{p_{i}(a*)}\right]$ by Equation (2a), a monotone increasing transformation of $\lambda +\mu \frac{p_{i}^{\prime }(a*)}{p_{i}(a*)}$ as U(·) is concave and consequently $U^{\prime }(\cdot )$ is a decreasing function. Next it is necessary to show that the likelihood ratio is increasingly affine in output (i.e., the output MED satisfies the monotone likelihood ratio property).

The output distribution has a probability function given by (1), so the likelihood ratio is:(3a)$$\frac{p_{i}^{\prime }(a)}{p_{i}}=-\frac{\partial A}{\partial a}-\frac{\partial B}{\partial a}x_{i}.$$

The multipliers *A* and *B* do not depend on *x*, so their derivatives do not depend on output either; then the likelihood ratio is affine in *x*. By using function *Z* defined above, we obtain:$$\frac{\partial A}{\partial a}=\frac{\left(\partial Z/\partial a\right)}{Z}=\frac{(-\partial B/\partial a)\sum_{i=1}^{N}x_{i}e^{-Bx_{i}}}{Z}=\frac{(-\partial B/\partial a)e^{A}\sum_{i=1}^{N}x_{i}e^{-A-Bx_{i}}}{Z}=-a(\partial B/\partial a).$$

So, the derivatives  and  have opposite signs because a is positive by assumption. Then, Equation (3a) can be expressed as:(3b)$$\frac{p_{i}^{\prime }(a)}{p_{i}(a)}=\frac{\partial A}{\partial a}\left[-1+\frac{x_{i}}{a}\right]$$

Now, we show that (∂A/∂a)<0 by showing that (∂B/∂a)<0. By taking the derivative of $p_{n}$ and using that (∂A/∂a)=−(∂B/∂a)a, we obtain:$$\partial p_{n}/\partial a=-(\partial /\partial a)e^{-A-Bx_{n}}=-[\partial A/\partial a+(\partial B/\partial a)x_{n}]p_{n}=-(\partial B/\partial a)(-a+x_{n})p_{n}.$$
The sum of all the derivatives $\partial p_{n}/\partial a$ (n=1,…,N) must be zero as, for any effort, the mass of probability remains equal to 1. Let us show ∂B/∂a<0. Assume ∂B/∂a>0. Given that E(X)=a holds by assumption, a higher effort a increases the probabilities corresponding to *n* low (*n* such that $x_{n}-a<0$), because $\partial p_{n}/\partial a=-(\partial B/\partial a)(x_{n}-a)p_{n}$ is then positive, and decreases the probabilities corresponding to *n* high (*n* such that $x_{n}-a>0$), because $\partial p_{n}/\partial a$ is then negative. This implies that (∂/∂a)E(X)<0, which is a contradiction because E(X)=a. Therefore, the hypothesis ∂B/∂a>0 results
in a contradiction and given that E(X)=a also rules out ∂B/∂a=0 by Equation (1), ∂B/∂a must be negative for any effort.

Consequently, the likelihood ratio $p_{i}^{\prime }(a)/p_{i}(a)$ is increasing affine in output as $\frac{p_{i}^{\prime }(a)}{p_{i}(a)}=\frac{\partial A}{\partial a}\left[-1+\frac{x_{i}}{a}\right]$ by Equation (3b) and ∂A/∂a>0. □

In the proof of Proposition 1, it was shown that the MED of the output satisfies the (commonly known as) monotone likelihood ratio property (MLRP henceforth). A direct consequence of the fact that $p_{i}(a)$ satisfies the MLRP is the following:

**Remark** **1.**
*Increasing the effort shifts the output distribution to the right in the sense of first-order stochastic dominance, a common assumption in agency theory. This property is stronger than*
(∂/∂a)E(X)>0
*and weaker than the MLRP.*


The optimal contract derived in Proposition 1 is curved in general, but affine in a transformed space. If, for example, U(·)=ln(·), then $s*(x)=\alpha +\beta (p_{i}^{\prime }(a*)/p_{i}(a*))$, with the likelihood ratio $p_{i}^{\prime }(a)/p_{i}(a)$ increasingly affine in output by Proposition 1; so, if the agent has logarithmic utility function, the second-best contract is affine in output. For the utility function is $U(\cdot )=2\sqrt{\cdot }$, the contract is $s*(x)={\left[\alpha +\beta (p_{i}^{\prime }(a*)/p_{i}(a*))\right]}^{2}$, and so the second-best contract is transformed-affine. Later in the paper in Section 4, we examine how the introduction of additional information into the problem (through additional constraints) may (or may not) change the affine pattern of the likelihood ratio.

### 3.2. Implications

Proposition 1 has some relevant implications in two cases, when the total output of the principal is the aggregate of individual output variables, and when the agent is risk-neutral.

Consider that the output *X* is the aggregate from the outcomes of *M* independent managerial projects. In the first situation, each project can either fail or succeed. The effort per project is a/M (i.e., increasing the agent’s effort a increases the probability of success of each and all projects), but the probabilities of success across projects (conditioned on effort a/M) are not necessarily equal. In the second one, the outcome of each project is measured by a (natural) number, X∈{0,1,2,3,…}; the effort per project is a/M, but in this second case, increasing the agent’s effort a increases the expected outcome of each project, $E(X_{i})=a/M$. In both cases, the output distribution is a MED (see the proof of the proposition below), and the corresponding probability functions are typically bell-shaped. The first case corresponds to a binomial distribution, used for example in [34] to model sales-force incentive contracts. In the second case, the output follows a negative binomial distribution.

**Proposition** **2.**(a) *If output X is defined as the number of successful projects among M independent (but not necessarily identically distributed) projects and*
E(X)=a, *the second-best contract is (after a monotone transformation) increasing affine in output*.(b) *If output X is obtained by aggregating the outcomes of M independent projects*, $X=X_{1}+\ldots +X_{M}$, with $X_{i=1,\ldots ,M}$
*being identical random variables defined on the support set {0, 1, 2, 3, …} and*
$E(X_{i})=a/M$, *the second-best contract is (after a monotone transformation) increasing affine in output. (The condition of independency is relaxed in* [35]).

**Proof.** In both cases, the FOA can be invoked by appealing to [22] (Cor. 1) or to [33] (Cor. 4), so the second-best contract is a monotone transformation of an affine function of the likelihood ratio.(a) The variable describing the failure or success of each project obeys a Bernoulli distribution. The sum of M independent Bernoulli distributions with different parameters follows an M-generalized binomial distribution. Among the M-generalized binomial distributions, a binomial distribution is the maximum-entropy one (see [36]), so according to (A3), the principal and agent will agree on that output follows a binomial distribution with E(X)=a. It must be noted that the probability of success in each project is not necessarily equal to *a/M*. The probability function of the binomial distribution with E(X)=a is equal to $p_{k}(a)=c{\left(a/M\right)}^{k}{\left(1-a/M\right)}^{M-k}$, where the integration constant c involves the binomial coefficients. The computation of the likelihood ratio for an output that follows a binomial distribution is straightforward:
$$\frac{p_{k}^{\prime }(a)}{p_{k}(a)}=-\frac{M}{M-a}+\left[\frac{1}{a}+\frac{1}{M-a}\right]k$$ increasingly affine in *k*.(b) By (A3), the distribution of X has maximum-entropy, which implies that each must have the maximum-entropy (as the projects are independent). The MED defined on the support {0, 1, 2, 3, …} is the geometric distribution, a particular case contemplated in Proposition 1. The sum of *M* independent geometric distributions with $E(X_{k})=a/M$ follows a negative binomial distribution with parameters *M* and *p* = 1 − *a/M* (so that E(X)=a). The probability function is (up to constants) equal to $\frac{a^{k}}{{\left(M+a\right)}^{N+k}}$, so the likelihood ratio is $\frac{p_{k}^{\prime }(a)}{p_{k}(a)}=-\frac{M}{M+a}+\left[\frac{1}{a}-\frac{1}{M+a}\right]k$, increasingly affine in output.□

Consider now that the agent is risk-neutral and designs the contract. Ref. [37] solves for the optimal agency contract when both the principal (investor) and the agent (entrepreneur) are risk-neutral, but the agent is constrained by limited liability. If the contract is restricted so that the compensation of the principal is non-decreasing with output (a condition justified to rule out sabotages), the optimal agency contract turns out to be of a debt type: the agent appropriates all the output and pays the principal a fixed interest on their investment as long as the payment is compatible with the agent’s limited liability constraint. In other words, the agent appropriates max[x-I, 0] and pays the principal min[x, I], with I the amount invested by the principal plus a fixed interest on that investment. Proposition 3 establishes that under the MEP, the optimal agency contracts with risk-neutral agents and limited liability will be debt-like contracts, without requiring an additional assumption on the contract shape.

**Proposition** **3.***Assume that the agent is risk-neutral and is constrained by limited liability; assume also that output follows any of the MED considered so far (in Propositions 1 and 2); the rest of assumptions as in (A2) and (A3). Then, the optimal agency contract is a debt-like contract*.

**Proof.** Ref. [38] shows that the optimal contract is non-decreasing with output (a debt-type contract) if and only if the hazard rate of the output distribution decreases with effort. If the MLRP holds, the hazard rate of the output distribution necessarily decreases with effort (see [38] (p. 227)). As the MEDs in Propositions 1 and 2 satisfy the MLRP, the optimal agency contracts correspond to debt-type contracts. □

If MLRP does not hold, the first-best solution may be attainable with a “live-or-die” contract (see [37,38]). The debt contract coming out of Proposition 3 is second-best and precludes the (unrealistic) “live-or-die” contracts, justifying debt contracts in fairly general conditions.

## 4. Extensions

In this section, we extend the model to include additional (to *E*(*X*) = *a*) information about the output distribution, which is now described by means of a continuous random variable, not necessarily restricted to adopt non-negative values. Then, we present additional results on the second-best agency contracts with continuous MEDs.

### 4.1. Generalized MED

If output is described by a continuous variable, the output distribution is defined on a continuous support. Given a continuous density function *f*(*x*), the entropy (or differential entropy) of the distribution is defined as:H:=−∫f(x)ln[f(x)]dx
with *X* denoting the random variable and x any realization of *X* (in other words, *X* represents the ex-ante output, and *x* an ex-post one. It must be noted that differential entropy is not necessarily positive; the reason is that, contrary to the probability function of a discrete random variable, the density function can be greater than one in some interval(s).The concept of entropy on continuous spaces entails some technical problems that are solved with the relative entropy (or Kullback–Leibler divergence), in other words, with the entropy measured with respect to a reference distribution. The relative entropy encompasses the differential entropy defined above as a particular case. The information provided by the ex-post output *x* is represented as follows: I(x):=−ln[f(x)], so H(X)≡E[I(x)]. Except when clarifications are needed, we refer to output *x* or *X* indistinctly. 

The principal and agent can share information on the output distribution additional to E(X)=a, mathematically represented by the equation ∫ϕ(x)f(x)dx=b, where *b* is a parameter and ϕ(x)≠x is a smooth function (by assumption). This additional information can be on another moment of the output distribution, which can (but not necessarily) depend on a through ϕ(x;a) or b(a). For example, if the principal and agent know that the effort of the agent can affect the expected output but the output variance is a constant independent of the agent’s decisions, then $\phi (x)={\left(x-a\right)}^{2}$ and $b=\sigma^{2}$ (or, alternatively, $\phi (x)=x^{2}$ and $b=\sigma^{2}+a^{2}$). If instead, a higher effort means that the agent manages a higher number of projects, which implies a higher expected output and a higher variance, the mathematical specification may correspond to $\phi (x)={\left(x-a\right)}^{2}$ and $b=\sigma^{2}a$. Constant *b* can be an observable or inferable quantity measurable from the market data or the economic environment. The ME distribution is now the solution to a problem that is an extension to (P2) in two ways: (i) the output distribution is defined in a continuous support (which could include negative values), and (ii) there is a third constraint that comes from the additional information shared by principal and agent as presented above:$$\underset{f(x)}{\max}H$$
subject to:∫f(x)dx=1
(P3)∫xf(x)dx=a
∫ϕ(x)f(x)dx=b.
The problem can be generalized to an arbitrary number of constraints (see [9] Park and Bera (2009, Equations (5) and (6))). Next, we characterize the solution to problem (P3).

**Lemma** **1.***The maximum-entropy density corresponding to (P3) is*:
$$f_{ME}(x)=(1/\Omega )\exp[-Cx-D\phi (x)]$$*where C and D denote the multipliers associated to the two last constraints in (P3) and the constant Ω ensures that the mass of probability is 1*: Ω=∫exp[−Cx−Dϕ(x)]dx. *If the output is non-negative and the information shared by the principal and agent is restricted to the first and second constraints (i.e., the trivial one and E(X) = a), then the MED is exponential, with density*$f_{ME}(x)=(1/a)\exp(-x/a)$*(which represents the continuous version of Equation**(1),* $p_{i}=e^{-A-Bx_{i}}$*).*

**Proof.** Equation (4) is obtained solving the Lagrangian optimization problem (P3) applying the calculus of variations (see e.g., [39]). If the third constraint is removed, then *D* = 0 and the distribution turns into an exponential function. □

Constant Ω plays the same role as *Z*(*B*) in the discrete model of Section 2. Except for the normalization constant, the effect of the third constraint on the density function consists in multiplying the exponential density by the factor exp[−Dϕ(x)]. This can make the density more concentrated around the modal value, which can be seen as the consequence of incorporating the additional information on the output distribution. As pointed out by [9] (p. 220), “in some sense, the shape of the resulting density has a close link with the “inverted” shape of [the constraint]”. Thus, the available information shapes the density of the output MED, which reflects the known economic determinants.

The variety of distributions comprised in Equation (4) comes from the variety of constraints that can be added to the optimization problem, typically involving the support and the moments of the distribution. A list of maximum-entropy densities that result from the solution of (P3) (i.e., compatible with the MEP) can be found in [9] Table 1). Some examples are the normal, the gamma, Weibull, Laplace, and many others. For example, $\phi (x)=x^{2}$ implies that *X* follows a normal distribution, and ϕ(x)=ln(x) implies that *X* follows a gamma distribution. More sophisticated distributions can also be obtained from the MEP. For example, [9] proposes a maximum-entropy ARCH model to describe the high leptokurtic behavior of stock returns (see also [40]). Importantly, the density in Equation (4) belongs to the “exponential family” (which implies that multipliers *C* and *D* are natural parameters for the ME density, with *x* and ϕ(x) the corresponding sufficient statistics). The result that all MEDs belong to the exponential family is particularly relevant for the agency problem because, as shown by [22] (Cor. 1) and [33] (Proposition 5), weak conditions suffice to validate the FOA when the output density belongs to the exponential family.

### 4.2. Agency Contracts

The following result provides sufficient conditions ensuring affine likelihood ratios (and consequently, transformed-affine second-best agency contracts) for the MEDs that solve problem (P3).

**Proposition** **4.***The likelihood ratio of the MED that solves (P3) is increasingly affine in output if any of the following conditions holds*: (a) *Output x is obtained by aggregation of M (i.i.d.) shocks*$x_{i}$*about which the only information available is*$E(x_{i})=a$, *i = 1, …, M*. (b) *The derivative*(∂/∂a)(Dϕ(x))*is zero*. 

**Proof**.(a) The only information available about the individual shocks is $E(x_{i})=a$, so each shock follows an exponential distribution. The sum of *M* (i.i.d.) exponential shocks follows an Erlang distribution, which is a particular case of the gamma. The density function is $f(x;a)=\frac{1}{a^{M}\Gamma (M)}x^{M-1}e^{-(M/a)x}$, with Γ(·) the gamma function. The likelihood ratio is $f_{a}/f=-M/a+(M/a^{2})x$, affine in output. This distribution satisfies the requirements for the FOA to hold, see [22] (Cor. 1), and recall that U[J(·)] has been assumed to be concave in (A1). (b) From Equation (4), we obtain the likelihood ratio:
$f_{a}/f=-(\partial \Omega /\partial a)\Omega^{-2}-(\partial C/\partial a)x-(\partial /\partial a)(D\phi (x))$. The last term is zero by hypothesis so the likelihood ratio is $f_{a}/f=-(\partial \Omega /\partial a)\Omega^{-2}-(\partial C/\partial a)x$. This ratio will be increasingly affine in *x* when the derivative ∂C/∂a<0 is negative. To show that ∂C/∂a<0, we followed similar steps to show that ∂B/∂a<0 in the proof of Proposition 1: $E(f_{a}/f)=0$ because f(x;a) integrates to one. Given that Ω and *C* do not depend on output *x* and that $f_{a}/f=-(\partial \Omega /\partial a)\Omega^{-2}-(\partial C/\partial a)x$, the likelihood ratio $f_{a}/f$ either increases or decreases with *x*. Let us see that $f_{a}/f$ decreasing with *x* entails a contradiction: consider the distribution function $F(x)\equiv \int_{}^{x}f(x;a)dx$; its partial derivative with respect to *a* is $F_{a}(x)\equiv \int_{}^{x}f_{a}(x;a)dx=\int_{}^{x}\frac{f_{a}(x;a)}{f(x;a)}f(x;a)dx$. The assumption that $f_{a}/f$ decreases with *x*, jointly with $E(f_{a}/f)=\int_{}^{}\frac{f_{a}(x;a)}{f(x;a)}f(x;a)dx=0$ implies $F_{a}(x)=\int_{}^{x}\frac{f_{a}(x;a)}{f(x;a)}f(x;a)dx>0$, which in turns implies that *E*(*X*) decreases with *a*: this contradicts the condition E(X)=a, so $f_{a}/f$ increases with *x* and therefore ∂C/∂a must be negative. □

Part (a) of the proposition resembles the result obtained for discrete outputs in Proposition 2(b), and is relevant because it justifies the commonly used gamma distribution in agency settings. Part (b) of the proposition holds for many known distributions; for the gamma distribution with density equal to $f(x)=\frac{1}{{\left(a/p\right)}^{p}\Gamma (p)}x^{p-1}e^{-(p/a)x}$, whose parameterization ensures that E(X)=a, the corresponding Lagrange multipliers associated to (P3) are *C* = *p/a* and *D* = 1 − *p*; for the normal distribution with density $f(x)=\frac{1}{\sqrt{2\pi \sigma^{2}}}e^{-\frac{{\left(x-a\right)}^{2}}{2\sigma^{2}}}$ (so E(X)=a and *Var*(*X*) = *σ^2^*), the corresponding Lagrange multipliers are $C=-a/\sigma^{2}$ and $D=1/(2\sigma^{2})$. For these two important distributions, the term Dϕ(x) is independent of a and then (∂/∂a)(Dϕ(x))=0.

In agency contracting, non-affine contracts cannot be excluded, neither theoretical nor empirically. Next, we provide conditions under which non-affine likelihood ratios arise in agency setting. Consider first that the output random variable follows a normal distribution with mean a and variance $\sigma^{2}a$, which corresponds to the MED with constraints E(X)=a and $E[{\left(X-a\right)}^{2}]=\sigma^{2}a$ (it must be recalled that, in order to validate the FOA when the likelihood ratio is unbounded-from-below, truncating the distribution support is needed; see [33] Chade and Swinkles (2020)). The specification intends to describe a situation where more effort entails the management of a larger number of projects and consequently a higher expected output and also a higher variance. The corresponding density function is $f(x;a)=\frac{1}{\sqrt{2\pi \sigma^{2}a}}e^{-\frac{{\left(x-a\right)}^{2}}{2\sigma^{2}a}}$, with a likelihood ratio equal to $f_{a}/f=-\frac{1}{2a}+\frac{x-a}{a\sigma^{2}}+\frac{1}{2a^{2}\sigma^{2}}{\left(x-a\right)}^{2}$. The likelihood ratio is convex in output, which could justify the use of convex compensation contracts when the agent’s utility function is logarithmic or “moderately more concave” than the logarithmic one. The question of justifying the FOA in this context is delicate: first, the likelihood ratio is U-shaped instead of monotonic (truncating the distribution support can circumvent this problem); second, in order to validate the FOA, the likelihood ratio is usually demanded to be weakly concave in output (see [22,33]).

Consider instead that the output is restricted to positive values, *x* > 0, and the available information is $E[\ln(X)]=\ln(a)-\sigma^{2}/2$ and $E[(\ln(X)-E(\ln(X)))]=\sigma^{2}$. The MED satisfying these two constraints is the log-normal; ln(*X*) is then a normal variable with parameters $\mu \equiv \ln(a)-\sigma^{2}/2$ and σ, and can be written as ln(*a*) plus a random term with normal distribution $N(-\sigma^{2}/2,\sigma )$. As usual, the expected output is $E(X)=\exp(\ln(a)-\sigma^{2}/2+\sigma^{2}/2)=a$. The corresponding density function is $f(x;a)=\frac{1}{\sigma x\sqrt{2\pi }}\exp\left[\frac{-{\left(\ln(x)-\ln(a)+\sigma^{2}/2\right)}^{2}}{2\sigma^{2}}\right]$, with *x* > 0, and the likelihood ratio is $f_{a}/f=\frac{1}{a\sigma^{2}}(\ln(x)-\ln(a)+\sigma^{2}/2)$, concave in output. The following proposition summarizes these results. Its application to agency contracting requires properly justifying the FOA.

**Proposition** **5.**(a) *If the shared information about the output distribution includes E(X) = a and*
$E[{\left(X-a\right)}^{2}]=\sigma^{2}a$, *the likelihood ratio is convex*.(b) *If the shared information about the output distribution includes*
$E[\ln(X)]=\ln(a)-\sigma^{2}/2$
*and*
$E[{\left(\ln(X)-E(\ln(X))\right)}^{2}]=\sigma^{2}$*, the likelihood ratio is concave.*

**Proof**. See the explanation above. □

Other functional forms of output-based compensation different from convex or concave ones can be justified by appealing to the MEP, for example, incentive schemes with a pattern first convex and then concave (see Section 5). 

### 4.3. Illustration

The following example illustrates the calculation of the second-best optimal agency contracts, and the effect of additional information on effort and welfare. Consider a gamma distribution with parameters *p* > 0 and *a/p* > 0. This distribution is the solution to (P3) with *E*(ln(*X*)) constrained to be a constant (expressed in terms of the parameters of distribution *X*, which implies that the third constraint in (P3) depends on effort a); if parameter *p* is an integer number, then output x can be interpreted as the aggregation of p independent projects. The density is $f(x)=\frac{1}{{\left(a/p\right)}^{p}\Gamma (p)}x^{p-1}e^{-(p/a)x}$, so the parameterization ensures that *E*(*X*) = *a*, and the likelihood ratio is $f_{a}/f=-p/a+(p/a^{2})x$. Assume that the net utility of the agent is $U(s,a)=2\sqrt{s}-a^{2}/2$ and their reservation utility is *R =* 1.2 (see (P2)).

Suppose first that the only information shared by the principal and agent is E(X)=a. By Lemma 1, the MED of output is the exponential distribution (which corresponds to a gamma distribution with *p* = 1). Applying the FOA (see Equation (2a,b)), we obtain that the second-best optimal agency contract is $s*(x)={\left(0.46+0.38x\right)}^{2}$; this contract induces the agent to optimally exert an effort *a** = 0.76, and the corresponding expected profit for the principal is *E*(*x* − *s**(*x*)) = 0.12. Consider now *p* = 1.5 (the density is then inversely U-shaped). The entropy of the gamma distribution is $H_{GAMMA}=p+\ln(a/p)+\ln(\Gamma (p))+(1-p)(\Gamma^{\prime }(p)/\Gamma (p))$, which increases with *a*, and fixing *a*, it reaches its maximum at *p* = 1 (exponential distribution). So, when output follows a gamma distribution, the information contained in the third constraint necessarily implies an ex-ante informational gain (as the third constraint reduces the output entropy). The optimal contract is now $s*(x)={\left(0.43+0.41x\right)}^{2}$, and we can observe that the incentives are now more powered than in the exponential case (*p =* 1); see Figure 1. The corresponding second-best effort is higher, *a* =* 0.82, and the principal’s expected profit is also higher, *E*(*x* − *s**(*x*)) = 0.15. In sum: in this particular example, the additional information on output leads to a smaller output entropy, higher effort, and higher profits. The generalization of these conclusions needs additional research.

## 5. Discussion

### 5.1. On the Properties of the Second-Best Contracts

Proposition 4(a) links the aggregation of performance measures to affine likelihood ratios. The MED is an exponential function when output is non-negative and the only prior information on the output distribution is *E*(*X*) = *a*. In production situations where the agent’s effort influences the outcome of a number of independent projects that contribute equally to the aggregate level of output (for example, the “effort” of the CEO influences the performance of several divisions that contribute to the total profits of the company), the MED distribution of the output aggregate will be distributed according to a gamma distribution. The second-best contract for every project (or division) is transformed-affine (by a similar result to Proposition 1), with identical slopes as the effect of effort is identical across projects (by assumption), so the contract on the aggregated output variable keeps the transformed-affine property. The gamma distribution is often used in agency research ([34,41,42]). Proposition 4(a) can also be related to the result on aggregation and linearity of [28], but here the aggregation is not over time but on parallel realizations of independent variables.

As shown in Proposition 4(b), condition (∂/∂a)(Dϕ(x))=0 ensures an affine likelihood ratio (i.e., a transformed-affine contract). It is sufficient for the condition to hold that the third constraint, ∫ϕ(x)f(x)dx=b, is independent of a, but this is not a necessary condition; for example, the gamma distribution is a MED whose third constraint, ∫ln(x)f(x)dx=const. involves effort a (see the example in Section 4.3). In sum, Proposition 4 gives sufficient conditions ensuring that the second-best optimal contract is increasingly affine in output in a transformed-space. In the particular case where the agent’s utility function is logarithmic, the second-best contract is affine, an incentive scheme commonly observed in practice.

On the other hand, Proposition 5 provides conditions under which the likelihood ratio is not affine in output, but rather convex or concave. This may explain the use of more sophisticated incentive contracts in managerial practice. For example, incentive contracts with stock options, where compensation is convex on the value of the performance variable (e.g., shares of the company); this kind of incentive can be optimal if the agent can modify the variance of the performance variable (as in Proposition 5(a)) and the principal (expectedly) benefits from more dispersed outputs, due, for example, to their risk-neutrality and limited liability. Besides, Proposition 5(b) suggests that (theoretical) second-best optimal contracts can also be concave. A pattern first convex and then concave, which resembles incentives based on performance standards, is also compatible with the MEP, for example, if the MED is the Laplace distribution (the corresponding contract can be seen as a particular case of the second-best schemes based on performance standards obtained in [43]). In sum, the MEP approach can accommodate empirical regularities observed in actual contracts.

In particular, the framework proposed justifies that observed contracts tend to be more detailed and sophisticated in mature and developed environments (in which trial-and-error, learning, and trust after repeated transactions play a role) than in environments where information is scarce. For example, sales forces are often paid by means of a fixed salary plus a percentage of sales as long as the sales exceed a pre-specified threshold ([41]), and a similar situation can be observed in public traded companies, where the managerial incentives include stock options; the performance standards together with a “cap” frequently used in managerial compensation contracts ([20]) would also reflect some learning by experience by managers and their employers.

### 5.2. Relation with Other Approaches to Agency

Next we related the MEP approach with alternative ways of handling situations where agency contracting takes place in conditions different from those considered in the standard case. The MEP approach to agency is related to Carroll’s analysis [44] and to ambiguity-aversion models (see references below). In Carroll’s model, the agent has information on the production possibilities set that the principal does not have, but for the actions known by the principal and agent, the output distributions are known (quantifiable uncertainty); the principal responds to the information advantage of the agent, offering the agent a compensation contract chosen under the worst-case criterion. Despite the profound differences in their starting points, the second-best contract obtained from the MEP in our basic case (Section 3) resembles the optimal contract in the worst-case-scenario agency model proposed in [44] (in which, if the agent is risk-averse, the second-best contract is affine after a utility-dependent transformation). Interestingly, in informational terms, the second-best transformed-affine contract obtained from the MEP is compatible with a choice under a worst-case-scenario (max-min equilibrium of the game against nature, see Section 2.3).

A more general approach to decision-making under uncertainty corresponds to ambiguity-aversion decision models (see [45] for a review and a critical assessment). This strand of the literature has been motivated as a response to the Ellsberg Paradox and similar violations of Savage’s axioms (see [46]), where the standard expected-utility model fails as criterion to rank alternatives with uncertain payoffs because the uninformed decision-maker cannot assign probabilities to states of nature. In ambiguity-aversion models, the decision-maker chooses the decision (or control) variable trying to maximize the corresponding expected utility, with a utility function that reflects their ambiguity-aversion; the mathematical specification of the expected utility can adopt different forms (see [47,48,49]). In contrast, the MEP imposes the condition of independence between probabilities and payoffs (Savage’s Postulate 4), moving from an initial situation of uncertainty with an unknown output distribution to one of risk (so, the decision-maker has no possibility of expressing aversion to ambiguity and can be qualified as “ambiguity-neutral”). Consequently, under the MEP approach, there is a unique, privileged distribution (the MED) that describes the environment uncertainty. Although each state of nature will have an associated monetary payoff, these payoffs do not intervene in the elicitation of the probability distribution. For this reason, the Ellsberg Paradox (and other experimental refutations of Savage’s postulates) remains incompatible with the MEP.

In the application of ambiguity-aversion to agency (see [27,50,51,52]) the decision problem of the agent is modified to incorporate ambiguity (via multiple probability distributions) and ambiguity-averse preferences (via their utility function). Then, the formulation of the agency problem under ambiguity-aversion demands that the principal and agent share a huge amount of information, and solving for the second-best contract requires considering the multiple incentive compatible constraints from the multiple prior distributions (see [51]). Instead, the information that the principal and agent share under the MEP approach is even less than the information presumed in the standard contracting situation, where both agree on the output distribution. In sum, a disadvantage of the MEP approach to agency lies in its incapability to justify some decision-making behaviors documented in the literature (see [46,53]). On the other hand, the simplicity of the MEP approach and its compliance with Savage’s axioms represent advantages with respect to other models.

One possible way to link the MEP approach with ambiguity-aversion models is through the concept of maximum-relative-entropy (MRE henceforth) or Kullback–Leibler divergence (see [54]). By appealing to the MRE principle, the choice of the distribution is carried out with respect to a reference distribution (or more generally, to a given background measure; for the Lebesgue measure, the MRE coincides with the differential entropy introduced in Section 4), which must not be interpreted necessarily as a prior distribution. The MRE principle could accommodate ambiguity-aversion models of the kind proposed by [49] as well as Bayesian updating ([26] (Section 8)).

## 6. Summary, Conclusions, and Further Research

How do we proceed in an agency-contracting situation under moral hazard where the probability distribution of output (conditioned to the agent’s effort) is unknown for both the principal and agent? The present study proposes the MEP as a framework that provides a rationale for how equally uninformed parties can converge to an agreed focal output distribution that fits the shared information consistently with principles established in information theory. The framework can be considered a first attempt to extend the MEP from individual decision-making under uncertainty to transaction situations between two parties under asymmetric and incomplete information. More specifically, the article shows how to embed the input provided by the agent in the problem that solves for the MED subject to the information shared by the principal and agent, and then it characterizes the second-best contracts that result from solving the agency contracting problem with the corresponding MEDs.

The results show that when the information available is the minimum required for the agency relation to make sense (i.e., the shared knowledge reduces to the agent’s effort driving the expected output), the second-best contract follows a transformed-affine increasing function of output. When additional information is incorporated to the problem, the approach can accommodate a long list of second-best contracts observed in business compensation settings, for example, stock options or capped bonus; the transformed-affine pattern of compensation remains under particular conditions, but more sophisticated contracts can also arise. Thus, the MEP approach to agency provides a unified framework based on the information available, not more nor less, which is able to accommodate many of the output distributions assumed in agency research and, consequently, many of the second-best contracts obtained by theory research as well as incentive contracts used in business settings.

The study opens a line of future research to explore the connection between the MEP approach to agency and ambiguity-aversion models through the concept of relative entropy. This extended notion of entropy can also be used to analyze the design of dynamic incentives under Bayesian updating of probability distributions, for example, in situations where information is periodically updated. Another line of future research could be to explore the implications for the contract design of additional information on other variables that share the same states of nature, for example, variables informative about rival firms that could justify the use of incentives based on relative performance. In a model of relative performance incentives with updating, the concept of transfer entropy, used in econometrics and financial time series analysis (see [10,55,56,57]) might be of interest, since the information on the relevant output variable could improve with observations of related variables.

## Figures and Tables

**Figure 1 entropy-23-00957-f001:**
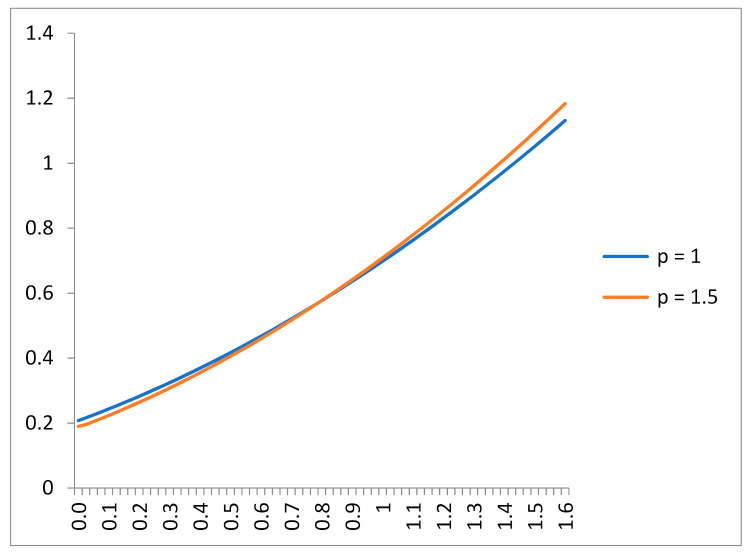
Figure exhibits the second-best optimal contracts of the illustration for the range of output values 0 < *X* < 1.6. The compensation of the agent follows a quadratic functional form $s*(x)={\left(\alpha +\beta x\right)}^{2}$, with *α* and *β* as positive constants. The contract turns out to be affine in output after a monotone transformation linked to the agent’s utility (in particular, the square root of the agent’s payoff). The blue line corresponds to the case *p* = 1; the corresponding output MED is exponential, obtained from (P3) with two constraints (the trivial one and *E*(*X*) = *a*). The red line corresponds to the case *p* = 1.5; the corresponding output MED is an (inverted U-shaped) gamma distribution, obtained from (P3), but in this case also constrained to *E*(ln*X*) = *const*. We can observe that in this example, the additional information (on *E*(ln*X*)) does not change the pattern of the second-best contract; Proposition 4 provides sufficient conditions for that to happen. In this example, the additional shared information leads to a second-best contract with more powered incentives: the fixed part of the incentives is lower and the slope of the incentives function is higher (*p* = 1.5); the corresponding contract leads to a higher agent’s effort *a** and a higher principal’s profit *E*(*x* − *s*(*x*)).

## Data Availability

The study does not report any data.

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
