# Peer review of "Agency Contracts under Maximum-Entropy"

_entropy, 2021, doi:10.3390/e23080957_

Round 1

Reviewer 1 Report

The authors study an application of maximum entropy principle in the context of agency contracts. Particularly, a principal hires an agent for a specific task, but without having the full information about the probability distribution of the output, which is a function of the agent's effort. For a risk neutral agent, the minimum information needed turns out to be the expected output for the principal. The authors go on to add the conditions for risk averse and risk taking agents. The manuscript is very thorough and treats the problem at hand from all conceivable angles. Therefore, I recommend its publication as is.

Reviewer 2 Report

The paper discusses agency contracts for the case of MEP. The paper is very technical and it is hard to read. The MEP is here only used to derive the relevant distribution (which is also called softmax). Otherwise, it is another type of optimization (P2), which uses MaxEnt distribution. In my opinion, the paper needs a major revision to make it more readable to the audience of the journal. It will be great to provide an example of an application. The authors compare their results with the other relevant results on this application. I suggest cutting the main text and focusing on the main aspects of the MEP in the model and what are the advantages/disadvantages.  

Round 2

Reviewer 2 Report

The manuscript has been improved. I suggest just a few minor points before the publication:

1) can the authors plot the solution of the illustrative example?

2) can authors make a short overview of information economics at the beginning? The authors claim that the principle of maximum entropy has been ignored, but as far I know, in econometrics, its version called softmax is used quite often. Can the authors describe this issue a bit closer in the introduction?

3) Note that many concepts based on entropy (e.g., transfer entropy) are used in econometrics and financial time series analysis. Is there also some relation to agency contracting?

4) Can you please change the style of the paper to the Entropy template? 
